psychology

COVID-19, polling data, public support, policy, attitude formation

**Author for correspondence:**
Colin M. G. Foad
e-mail: cfoad@ed.ac.uk

†Present address: School of Social and Political Science, University of Edinburgh, 15a George Square, Edinburgh EH8 9LD, UK.

# The limitations of polling data in understanding public support for COVID-19 lockdown policies

Colin M. G. Foad[1,†], Lorraine Whitmarsh[2], Paul H. P. Hanel[3] and Geoffrey Haddock[1]

[1]Department of Psychology, Cardiff University, Cardiff CF10 3AT, UK
[2]Department of Psychology, University of Bath, Bath, UK
[3]Department of Psychology, University of Essex, Colchester, Essex, UK

CMGF, 0000-0003-0423-2848

Opinion polls regarding policies designed to tackle COVID-19 have shown public support has remained high throughout the first year of the pandemic in most places around the world. However, there is a risk that headline support over-simplifies people's views. We carried out a two-wave survey with six-month interval on a public sample ($N = 212$) in the UK, examining the factors that underpin lockdown policy support. We find that the majority of people support most public health measures introduced, but that they also see significant side effects of these policies, and that they consider many of these side effects as unacceptable in a cost–benefit analysis. We also find that people judged the threat of COVID-19 via the magnitude of the policy response, and that they do not use their perception of the personal threat to themselves or close others to guide their support for policy. Polling data only offer one simple perspective and do not illustrate the ambivalence many people feel around lockdown policies. There is also a meaningful risk of public opinion and government policy forming a symbiotic relationship, which impacts upon how effectively such policies are implemented both now, and in relation to future threats.

## 1. Introduction

The COVID-19 pandemic has seen governments across the world implement previously unprecedented policies, significantly limiting the basic freedoms of their populations. A central feature of the pandemic has been widespread support for these policies throughout the general public and across political divides [1,2], with this support regularly reported in the press and on social

media. International polling data show that this support is partially dependent upon type of policy, country and date; with some policies becoming less supported over time (e.g. cancelling non-COVID health services), others maintaining consistent support (e.g. banning large events) and some rising over time (e.g. quarantining international travellers) [3]. In the UK, support for the third national lockdown remained high in mid-January 2021, with the public supporting even stricter measures than those enforced by the government [4]. While other polling data subsequently showed that many individuals took that lockdown less seriously and found it more difficult [5], the government's decision to delay the final step of lockdown easing still carried majority public support in June 2021 [6].

While we, therefore, have broad headline information about *what* people think, such polling data have much less utility in understanding *how* people reach their position. In more typical contexts (e.g. congestion charges in city centres), public acceptability of policy is predicted by the policy's perceived efficacy (does it work?), personal outcome expectancies (will it be good or bad for me?) and perceived fairness (e.g. will this affect some people more than others?) [7]. However, the urgent nature of the pandemic forced the public to assess the acceptability and efficacy of exceptional policies in a very short space of time. This article outlines relevant fundamental psychological perspectives before examining how such theory, aligned with our data, provides a more comprehensive understanding of public opinion than available through standard polling.

Before describing relevant theoretical positions, it is important to outline our overarching perspective. We wanted to understand further what underpins consistent polling data that show high support for lockdown policies. In particular, we sought to determine the information that people use to assess the threat of COVID-19, as well as how they see the cost–benefit trade-off of lockdown policies. Typically, the generally high levels of support for lockdown policies reported by polling companies have been contextualized as representing the view that the public perceives the intended benefits of lockdown as outweighing the costs of its side effects. However, these data only examine support from one angle, and cannot capture a range of views regarding the impacts of lockdown. Additionally, we suggest that polling support for lockdown has often been interpreted as evidence that this is what people consider appropriate from their personal perspective. We offer an alternative hypothesis: that asking about support for lockdown in isolation does not adequately capture the concerns people also have about lockdown side effects, and that when people are asked to judge the threat of COVID-19, they base their judgement on the policy response (i.e. shift from 'normality' to lockdown), rather than on their assessment of the threat to their personal circumstances.

In order to test this hypothesis, we examined three important and related questions. First, does support for lockdown policies also mean an awareness that side effects (i) exist, and (ii) are acceptable? Second, do people use the magnitude of the response to guide their support for lockdown? Third, do people use the personal threat to themselves and close others, or a more general assessment of the threat, to guide their support for lockdown?

All of these questions are linked with the concept of attitude. The study of interconnections among an individual's attitudes (i.e. opinions) has an extensive history within behavioural science. Cognitive dissonance theory [8] posits that individuals seek consistency among their attitudes, cognitions and behaviours. Similarly, research on hypocrisy shows how behaviour change is enhanced by highlighting to an individual a discrepancy between a public action they performed promoting an attitude (e.g. publicly stating the importance of doing X) and a behaviour they performed contradicting that attitude (e.g. recalling having done the opposite of X) [9]. To achieve a dissonant state, therefore, requires simultaneously making more than one attitude or behaviour salient. If only one perspective is examined in isolation, people can support a particular action without having to address any of the possible negative effects of that action, as little or no dissonance will be elicited.

More generalist models of attitude have considered how interconnections among beliefs can have causal influences on each other in determining an individual's overall attitude [10]. Importantly, people can simultaneously feel both positive and negative about an attitude object, and accessing this *ambivalence* requires going beyond simply examining general support [11]. As applied to the context of our research, we, therefore, needed to examine whether support for lockdown policies was consonant with an awareness of the side effects, and how individuals construe the trade-off between lockdown side effects and policy benefits. Crucially, all these theoretical positions allow room for inconsistencies within networks, and mapping these inconsistencies within the context of COVID-19 is only possible by assessing a position from multiple perspectives [12].

People also use heuristics to guide their judgements and opinions, including cues such as the views of elites [13], as well as inferring what they think by observing their own actions [14]. Furthermore, heuristic

cues can work at the collective level, formed via 'availability cascades', where informational cues (setting our beliefs in line with our perceptions of the beliefs of others), and reputational pressures (avoiding censure by following the beliefs of others) combine to allow collective beliefs to form [15]. In the face of an unprecedented and urgent crisis like the COVID-19 pandemic, cues and availability heuristics such as these are likely to be very powerful. We suggest that one strong cue people used to judge the severity of the problem at the start of the pandemic was the policy response itself. We, therefore, needed to examine whether people felt that they had used lockdown policies as a piece of information to help determine the perceived severity of the threat, and how this judgement came to be associated with support for such policies.

Additionally, construal level theory shows how attitudes can form at different psychological distances and levels of abstractness, dependent upon the situational context [16]. Through this lens, we can examine whether polling questions about COVID-19 tap what people think is appropriate for *their* protection (close and concrete) or what people think the *country and/or world* needs (distant and abstract). If people have assessed the threat of COVID-19 at a psychologically close and concrete level, we would expect their concerns about the impact on them and close others to be the stronger predictor of their support for lockdown policies. However, if people have assessed the threat at a more psychologically distant and abstract level, then we would expect a general threat assessment to be the stronger predictor of lockdown policy support. Because the pandemic is, by its very nature, a global problem, it seems plausible to suggest that general level construals are going to be highly important for people in their evaluation of the threat resulting from COVID-19.

Together, these literatures suggest that while headline information can indicate simple support for a particular action, it cannot demonstrate any coexisting attitudes which may reflect tensions with such support. For example, one could expect that people would wish for more hospitals and schools, while separately people may also state a desire for paying lower taxes. Even though people are likely to have some awareness about the inherent trade-offs embedded within different policies, those trade-offs may not neatly map onto their support for each policy when questioned in isolation. Similarly, with COVID-19, an individual's level of support for lockdown policies cannot directly illustrate their level of concern regarding the (negative) side effects of such policies. It is, therefore, necessary to examine individual attitudes from multiple perspectives, to understand how people perceive the trade-offs between tackling COVID-19 and other personal and social needs.

Given the interconnectivity among public opinion, media and politics [17,18], it is crucial that we understand these detailed perspectives in greater depth. For example, if the public uses the magnitude of a policy response (e.g. unprecedented lockdown) to guide their judgement for the suitability of such policies, and the government use public opinion to inform their views regarding the acceptable level of restrictions to implement, then there is a strong potential to maintain symbiosis between government policy and public opinion. Interestingly, baseline government approval in Sweden (which chose to use less restrictive policies) has tracked similar to other Western European nations throughout the pandemic [19], reinforcing the point that comparable levels of public support can be reached from very different approaches.

To understand the foundations underpinning public support for lockdown policies, we therefore need to examine how people view the side effects of such policies, to what extent people use logical heuristics to guide their judgements (e.g. 'COVID-19 is serious because we don't use lockdowns for other illnesses'), and whether individuals assess the threat *personally* (e.g. 'am I or close others likely to come to harm?') or *generally* (e.g. 'the virus is a threat to the country'). By assessing people's views at two time points in our research, we are also able to test whether these processes are robust across time.

# 2. Methods

## 2.1. Participants

The data for this article were collected as part of a larger project assessing attitudes towards COVID-19 from people living in the UK. The data were collected in two phases. A power analysis revealed that we would need 194 participants to detect a small-to-medium effect size of $r = 0.20$ with a power of 0.80. In phase 1 (30 June 2020), the sample comprised 212 participants (113 female, 92 male, 3 other, 1 prefer not to say, 3 missing; $M_{age} = 35.9$, s.d.$_{age} = 13.6$). Of these, 150 participants returned for the second phase (14 December 2020). Participants were recruited using Prolific (www.prolific.co) and paid for their time.

**Table 1.** Descriptive and reliability statistics for each scale.

| | phase 1 | | | phase 2 | | |
|---|---|---|---|---|---|---|
| | M | s.d. | α | M | s.d. | α |
| threat assessment: personal | −1.65 | 1.94 | 0.76 | −1.76 | 1.98 | 0.75 |
| threat assessment: general | 1.62 | 2.11 | 0.80 | 1.27 | 2.45 | 0.87 |
| judging threat via response | 3.03 | 1.62 | 0.79 | 2.58 | 2.09 | 0.88 |
| lockdown policy initial support: higher agreement | 3.96 | 1.51 | 0.90 | 3.23 | 1.95 | 0.92 |
| lockdown policy initial support: lower agreement | 1.51 | 2.67 | 0.91 | 0.20 | 2.78 | 0.92 |
| lockdown policy future support: facemasks | 1.33 | 2.44 | 0.87 | 1.65 | 2.33 | 0.84 |
| lockdown policy future support: public restrictions | 1.43 | 2.37 | 0.62 | 1.62 | 2.47 | 0.65 |
| side-effect prevalence: individual | 2.53 | 1.44 | 0.85 | 2.79 | 1.60 | 0.92 |
| side-effect prevalence: social relations | 1.12 | 1.86 | 0.73 | 1.14 | 1.83 | 0.72 |
| side-effect prevalence: society | 3.10 | 1.43 | 0.77 | 3.25 | 1.56 | 0.87 |
| side-effect trade-off: individual | −1.23 | 2.30 | 0.94 | −1.17 | 2.29 | 0.93 |
| side-effect trade-off: social relations | 2.60 | 2.62 | 0.92 | 2.34 | 2.81 | 0.91 |
| side-effect trade-off: society | −1.01 | 2.23 | 0.79 | −1.49 | 2.33 | 0.88 |

## 2.2. Procedure and design

The measures reported below were designed specifically to address the question of how people formed their attitudes towards lockdown policies and have not been published in any form anywhere else. Future articles are planned to outline other psychological processes of interest from these datasets. A full list of items collected for each phase is available (see OSF link in Data accessibility). Participants completed both phases of data collection on Qualtrics. Participants were presented with a range of measures assessing attitudes towards aspects of COVID-19, individual differences, demographics and other measures unrelated to COVID-19. As long as participants completed at least half the measures, they were included in the analyses (six participants failed to complete half the measures in phase 1, and six participants failed to complete half the measures in phase 2). Approval for data collection was obtained from the local ethics committee.

## 2.3. Measures

The project captured a range of data examining COVID-19 attitudes, but we focus here on the relevant variables for this article. Table 1 outlines the mean, standard deviation and reliability for each measure at each time point. All items were measured on a scale from −5 (depending on item wording: *strongly disagree; strongly oppose; costs of side effect much worse than benefits of lockdown*) to +5 (*strongly agree; strongly support; benefits of lockdown much greater than costs of side effect*).

To reduce items into reliable scales, we adopted the same process for all measures. First, we entered the items for each measure into a principal components analysis (PCA), with Varimax rotation used when multiple components were present. We retained any components with an Eigenvalue greater than 1 and excluded any remaining items. Furthermore, to be retained, all items required a component loading of at least 0.60, and no cross-loading with other components greater than 0.50. The PCA analyses showing all retained and excluded items are available in electronic supplementary material (tables S1–S5), as well as the question framing and all scale anchor points. All measures reported below were designed for the purposes of this research, although they were informed by items being used by polling companies and government policies at the time of data collection. All the PCAs were run on phase 1 data and all the scales developed then showed good reliability at both phase 1 and phase 2 ( table 1).

### 2.3.1. COVID-19 threat assessment (personal and general)

Two types of threat assessment were measured: *personal threat* measured the perceived threat to themselves, friends and family (e.g. I think it is likely a close family member will die from COVID-19

at some point in the future), and *general threat* measured the perceived threat at a broader level (e.g. I think the number of deaths directly caused by COVID-19 is a massive threat to this country).

### 2.3.2. Judging COVID-19 threat via response

Five items were developed to assess whether participants had an awareness that they might be using the policy response to guide their judgement of the size of the threat (e.g. I know that COVID-19 is a serious issue because we do not put lockdowns in place for other illnesses).

### 2.3.3. Support for initial COVID-19 restrictions

To assess participants' initial lockdown support, several aspects of the first lockdown policy, differing in ease of adherence and, therefore, likely agreement (e.g. easier: encouraging handwashing; harder: restricting visitor access to hospitals for terminally ill patients), were presented, and participants stated how much they supported each one at its time of inception. The PCA on these items revealed two components. The first contained items attracting high levels of agreement (e.g. closing all pubs and restaurants), and the second contained items attracting lower levels of agreement (e.g. restricting attendance at funerals).

### 2.3.4. Support for future COVID-19 restrictions

To assess participants' support for further restrictions in the future, a range of possible policies were presented examining actions such as facemask usage (e.g. make facemasks compulsory for young children over 3 years old on public transport) and public restrictions (e.g. ban public protests).

### 2.3.5. Understanding of policy side effects

To assess the extent participants thought there were consequent side effects of lockdown policies, a series of side effects were presented which captured participants' perceptions of the likelihood of impact upon individuals (e.g. increase in mental health problems), social relations (e.g. an increase in people fearing being judged by others in public) and on society (e.g. increased deaths from non-COVID-19 illnesses because of greater reluctance to attend hospital).

### 2.3.6. Policy side effects trade-offs

To assess how people saw the trade-off between the benefits of lockdown policies and the costs of the side effect, each side effect was presented again, and participants rated the cost–benefit analysis of each one.

## 3. Results

Consistent with previous polling data, figure 1 shows there was initial support for each aspect of lockdown policy in phase 1 (one-sample $t$-tests versus zero baseline; all $ts > 6.10$, all $ps < 0.001$) and for all policies other than restricting visitor access to hospitals in phase 2 (one-sample $t$-tests versus zero baseline; all $ts > 2.26$, all $ps < 0.03$), though initial support had dropped by phase 2 (paired-sample $t$-tests; all $ts > 3.19$, all $ps < 0.003$). These data align with polling [1,2,5] insofar as they represent majority support for lockdown policies, albeit with some attenuation over time.

We then assessed the extent to which participants perceived the likelihood of side effects from these lockdown policies. Figure 2 shows that for each category of side effect and at both time points, participants saw the side effects as having occurred because of lockdown policies (one-sample $t$-tests versus zero baseline; all $ts > 7.67$, all $ps < 0.001$). Next, having established the level of support for policies, and the perceived likelihood of side effects, we examined whether participants perceived the trade-offs of these side effects as acceptable. Figure 3 shows that participants perceived the trade-off upon social relations as acceptable at both time points (one-sample $t$-tests versus zero baseline; $ts > 10.15$, $ps < 0.001$), but saw the trade-offs as unacceptable for their impacts upon individuals and society (all $ts < -6.26$, all $ps < 0.001$). Taken together, these findings show that participants consistently support lockdown policies, believe that these policies have significant side effects, and that most of these side effects are unacceptable as a trade-off in terms of the damage they cause.

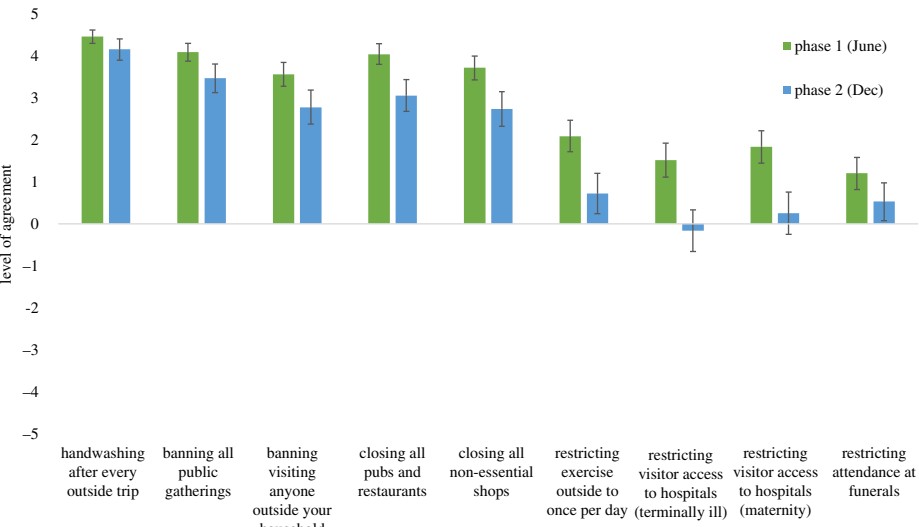

**Figure 1.** Lockdown policy initial support for phase 1 (June) and phase 2 (December) (error bars indicate 95% CIs).

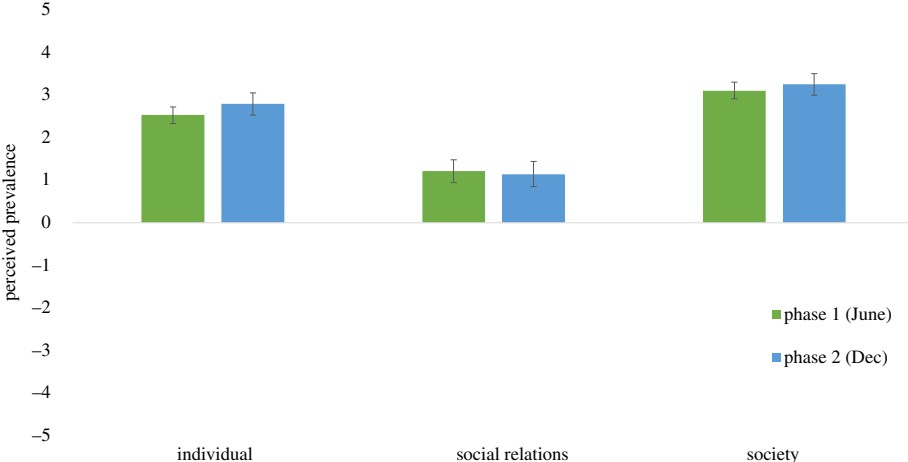

**Figure 2.** Perceived prevalence of side effects from lockdown policies (error bars indicate 95% CIs).

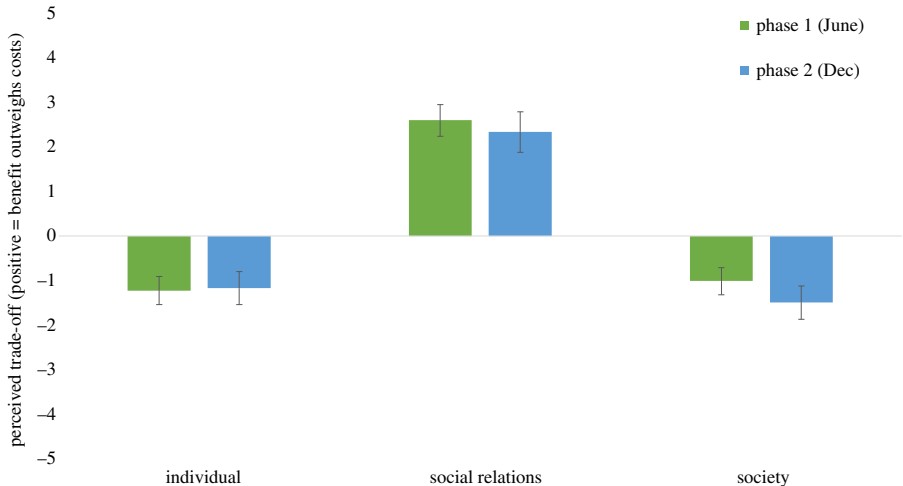

**Figure 3.** Perceived trade-offs of lockdown policies (error bars indicate 95% CIs).

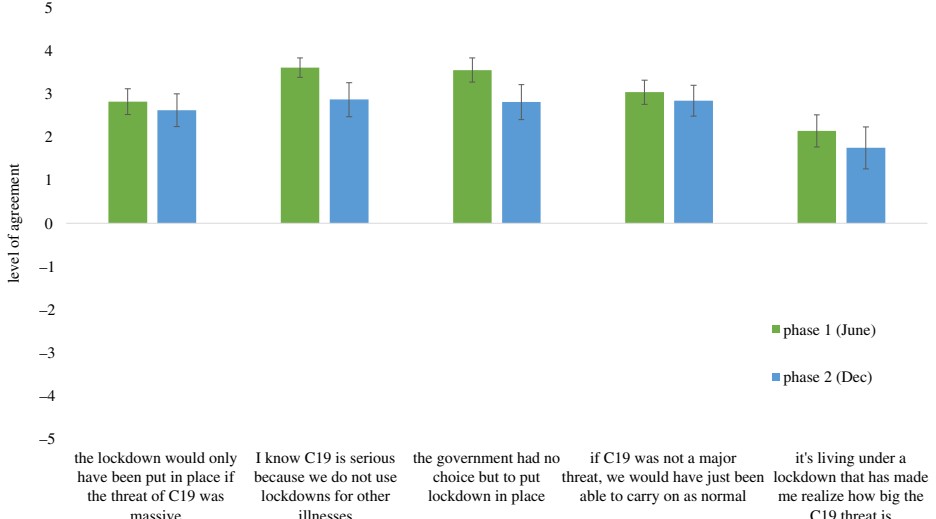

**Figure 4.** Judging the threat of COVID-19 via the lockdown policy response (error bars indicate 95% CIs).

Regarding our second question, figure 4 shows levels of agreement with the items that assessed whether or not participants used the policy response to guide their judgement of the threat of COVID-19. In both phases, participants showed very strong agreement with these items (one-sample t-tests versus zero baseline; all $ts > 7.10$, all $ps < 0.001$). Next, we examined whether this level of agreement (with the items combined into one scale) correlated with support for lockdown policies, which it did both at the point of inception (phase 1: higher agreement $r = 0.51$, $p < 0.001$, lower agreement $r = 0.34$, $p < 0.001$; phase 2: higher agreement $r = 0.70$, $p < 0.001$, lower agreement $r = 0.42$, $p < 0.001$), and for future support (phase 1: public restrictions; $r = 0.43$, $p < 0.001$; facemask policies, $r = 0.39$, $p < 0.001$; phase 2: public restrictions; $r = 0.58$, $p < 0.001$; facemask policies, $r = 0.64$, $p < 0.001$). While the cross-sectional nature of these data mean we cannot confirm direct causality between these variables, the results strongly support our hypothesis that participants used the policy response to guide their assessment of the threat, and accordingly their support for lockdown.

To test our third question, we examined the basis upon which people formed their lockdown attitudes. We used multiple regression to simultaneously predict whether participants' initial support for lockdown was predicted more by their assessment of the personal threat to themselves and close others, or by their assessment of the threat at a more general level (table 2). The analysis at phase 1 revealed personal threat did not predict initial support for lockdown policy but general threat did. This result was replicated at phase 2. Thus, at both time points, initial support for lockdown was predicted by general threat about COVID-19, not personal threat. We then tested whether this pattern was also found for participants' support for future policy. For phase 1, personal threat did not predict support for future public restrictions or facemask policies, but general threat predicted both these variables. Similarly, for phase 2, we found significant effects only for general threat. Together, these results show that people's support for lockdown policy, both at the point of inception and looking ahead to the future, was not predicted by their perception of the personal threat of COVID-19, but was strongly predicted by their general perception of the threat.

## 4. Discussion

Our findings offer three novel and important insights. First, when taken in isolation, participants strongly support lockdown policies, aligning with typical polling data. However, further measures assessing awareness and acceptability of side effects caused by those policies suggest a strong underlying ambivalence in relation to those headline figures. Overall, participants support lockdown, state that side effects occur and are then conflicted on whether or not those side effects are an acceptable trade-off (dependent upon the nature of the side effect). Second, participants state that they use the lockdown policy to infer their assessment of the threat, and the more participants do this, the more they support lockdown. Third, participants use their general assessment of the threat, but not their personal concerns, to guide their support for lockdown.

**Table 2.** Regression analyses comparing personal and general threat on support for lockdown policies (initial and future) at each phase.

| | initial support (higher agreement) | | | initial support (lower agreement) | | |
|---|---|---|---|---|---|---|
| | $B$ | s.e. | $p$-value | $B$ | s.e. | $p$-value |
| **phase 1** | | | | | | |
| personal threat | −0.10 | 0.05 | 0.065 | 0.00 | 0.11 | 0.997 |
| general threat | 0.36 | 0.05 | <0.001 | 0.23 | 0.10 | 0.020 |
| model summary | $R^2 = 0.20$, $F_{2,206} = 26.33$, $p < 0.001$ | | | $R^2 = 0.03$, $F_{2,206} = 3.44$, $p = 0.034$ | | |
| **phase 2** | | | | | | |
| personal threat | −0.14 | 0.08 | 0.063 | 0.12 | 0.13 | 0.335 |
| general threat | 0.53 | 0.06 | <0.001 | 0.25 | 0.10 | 0.020 |
| model summary | $R^2 = 0.37$, $F_{2,146} = 41.36$, $p < 0.001$ | | | $R^2 = 0.07$, $F_{2,146} = 5.59$, $p = 0.005$ | | |
| | future support (public restrictions) | | | future support (facemasks) | | |
| | $B$ | s.e. | $p$-value | $B$ | s.e. | $p$-value |
| **phase 1** | | | | | | |
| personal threat | 0.01 | 0.09 | 0.928 | 0.02 | 0.09 | 0.785 |
| general threat | 0.41 | 0.08 | <0.001 | 0.49 | 0.08 | <0.001 |
| model summary | $R^2 = 0.14$, $F_{2,206} = 16.48$, $p < 0.001$ | | | $R^2 = 0.19$, $F_{2,206} = 24.09$, $p < 0.001$ | | |
| **phase 2** | | | | | | |
| personal threat | 0.02 | 0.11 | 0.819 | −0.08 | 0.09 | 0.362 |
| general threat | 0.44 | 0.09 | <0.001 | 0.59 | 0.07 | <0.001 |
| model summary | $R^2 = 0.20$, $F_{2,146} = 18.34$, $p < 0.001$ | | | $R^2 = 0.36$, $F_{2,146} = 39.59$, $p < 0.001$ | | |

Polling data are regularly cited by influential sources within traditional and social media, and have a relatively long history of impact [20]. Recently, the link between social media content and journalistic representation of public opinion has become of particular interest [21]. It is thus important that we understand the complexities underlying apparently simple polling data, particularly if they are being used to infer unidimensional support or opposition for specific policies. This is crucial during the COVID-19 pandemic, which has elicited a prolonged period of unprecedented levels of news coverage for one issue. Additionally, consistent evidence shows the impact of various normative influences upon attitudes [15,22] and in relation to specific COVID-19 intentions such as the willingness to be vaccinated [23]. Accordingly, any oversimplified or non-representative presentation of public opinion risks reaffirming a normative judgement that, in reality, is likely to be much more nuanced than the conclusion being published, an issue magnified by the polarization of media coverage [24]. Furthermore, polling data have been shown to be susceptible to socially desirable responding [25], which aligns with theory on availability cascades [15], and could also contribute to exaggerating the true level of support for lockdown policies, particularly given the apparent moralization of the issue [26].

Aside from the issue of oversimplified evaluations of polling data, our findings also impact upon five further practical aspects of how future public health policy is formed for COVID-19 and other threats. First, unlike the advice given to the UK government's Scientific Advisory Group for Emergencies (SAGE) [27], our findings suggest that directly elevating people's perception of personal threat is not necessary for them to be persuaded to support lockdown. People typically judged the threat of COVID-19 at a general level, rather than because of the personal threat to them and their loved ones. Second, the actual efficacy of lockdown policies is a live debate and new policies are currently being considered to manage the future risks of COVID-19 (e.g. vaccine certification). If we are to avoid further misunderstanding of public opinion, then our findings should be taken into consideration when balancing the inherent trade-offs in such policies. Third, understanding that people are ambivalent about lockdown policies is important in countering the polarization that has appeared in academic discourse [28], as we need to communicate more of the complexity and nuance in people's

positions. This will be particularly important when the intricate details of where lockdown policies were and were not used effectively emerge in the future. It is likely that the initial strong support at the start of the pandemic will have led to the following inevitably uncertain information about lockdowns being interpreted through these strong lenses, further contributing to the polarization of public opinion; a process regularly found in other political issues, independent of individuals' scientific literacy [29,30]. Fourth, government policies and associated media coverage provide social norms and expectations for people about what is acceptable when tackling a global crisis [31], which is particularly likely to drive public opinion in an urgent context. If governments want the public to feel confident about the release of lockdown restrictions, then they need to clearly communicate why those restrictions are no longer necessary, nor likely to return in future. Fifth, our findings indicate people judged the threat of COVID-19 from the response, which in this case was a huge and sudden contrast between normality and lockdown. It is difficult to see how that contrast could be repeated in the short–medium term, which indicates the public may treat future public health threats relatively less seriously, independent of their actual severity.

The current findings are also relevant to findings for public health issues outside of COVID-19, where interdependence between public support and policy action has been an impediment to action on both sides. As one example, ambitious climate change policy was stalled for years because policymakers lacked a perceived social mandate for action [18], while sections of the public, conversely, presumed the risk of climate change was low because policymakers were not taking more radical steps to address it [32]. This tendency for public and the governments to displace responsibility for addressing collective problems like climate change has been termed a 'governance trap' [33] which can be overcome through strengthening democratic processes (e.g. deliberative democracy) [34]. In sum, there is much to be learned about how public opinion and government policy inform one another.

Before concluding, we note some limitations to our findings and consider future directions that could address them. First, most of our measures were developed in response to the pandemic and therefore, while informed by relevant attitudinal theory, it would be valuable to extend our research with additional qualitative and quantitative work which seeks to understand further how people formed their attitudes towards lockdown policies. Our findings offer a timely insight into how people felt at the time, but such work could help clarify any ambiguity in examining how people judge this and other threats via policy responses, how people interpret threat at different levels of construal [16], and how people quantify comparable risks [35]. Similarly, our data suggest that it is much easier to support lockdown in general when it is framed as the solution to the problem, than when the potential harms of the policy are considered. However, further work is needed if we are to more accurately quantify exactly where public tolerance lies in terms of trade-offs between COVID-19 risks, lockdown risks and also the positive aspects of normal social interaction.

Second, across the six-month period, many policies were constantly changing. Our measures were designed to examine attitudes independent of current policy, but longitudinal data could further test the relationship between the introduction of a policy and support for that policy, both within and outside the COVID-19 context. Third, while we used a public sample from a commonly used participant panel (Prolific), which offers significant heterogeneity in terms of participant gender, age, education and socio-economic status, the sample does have some political skew (towards left-wing and remain-voting participants). Nonetheless, we still had good sample sizes from centrists, right-wing and leave-voting participants, and did not find any strong relationships between political viewpoint and any of our measures (see electronic supplementary material, table S6), so we are confident the mechanisms we identify would translate into a more typically representative sample. However, given the comparisons with polling data, we do acknowledge this difference in sampling methodology.

Two additional theoretical future directions are worth considering in the light of our findings. First, we report a clear dissonance between overall support for lockdown and the cost–benefit analyses of associated policies. It seems reasonable to suggest that people find it easier to support lockdown policies within a positive frame (saving lives) than within a negative frame (causing harm to children). It would be interesting to examine further whether making both the policy and its side effects salient simultaneously arouses dissonance [8], and, if so, the potential for consequent attitudinal and behavioural change. Second, extant research on side effects shows that different types of side effect attract different attributions at the level of the individual [36]. It would be valuable to extend our understanding of how the side effects of lockdown policies are seen in terms of causation, and how this might impact upon people's tolerance for such effects.

Taken together, the findings from our study suggest that any presentation of polling data as evidence for strong public support for COVID-19 lockdown policy needs to be placed in a much richer context to

more accurately represent the perceived trade-offs in balancing public health risks with the broader costs of imposing lockdown policies. Without such a wider consideration, there is a real risk of overestimating the extent to which the public are coping with such restrictions, by implying that headline support for such policies indicates an underlying coherent attitude network towards associated impacts. Perhaps most crucially, we risk facing an in-built symbiosis between government policy and public support. Indeed, in effect we face an inherent triangulation, of attitudes towards the threat, the size of the policy response and the perceived solution to the threat (i.e. lockdowns). If, as we theorize, people have judged the threat of COVID-19 both at a general construal level and from the policy response, then we risk creating a self-fulfilling prophecy, where the public presume such measures are necessary from the very existence of such measures, and the government and the media presume such measures are desired by the public. It is hard to quantify many of the harms caused by COVID-19 and lockdowns, which is precisely why it is so important to understand the complexities underlying public opinion; otherwise, the aforementioned symbiosis may make each lockdown policy seem completely unavoidable, rather than the difficult choice made by a democratically elected government that it actually is. Given the widely accepted point that lockdown policies exacerbate many interconnected forms of inequality (e.g. gender, race, housing, employment, financial, health), it is, therefore, vital that none of us become overly relaxed in how long lockdown policies are maintained, relying upon headline polling support as comfort or justification.

Ethics. Ethics approval was granted by Cardiff University Ethics Committee (ref: EC.20.05.12.6034R) before data collection commenced. Informed consent was provided at the start of the study by all participants. Participants were debriefed and thanked at the end of each wave.

Data accessibility. The raw and modified datasets, complete sets of items used in both phases and the syntax files used for analyses are available (on OSF) at: https://osf.io/7ph4f/?view_only=27c3e7875ae7483d920910abb6aaca25. The study was not preregistered.

The data are provided in electronic supplementary material [37].

Authors' contributions. C.M.G.F.: literature search, study design, study coding, data collection, data analysis, data interpretation, writing. L.W.: literature search, study design, data interpretation, writing. P.H.P.H.: data analysis, data interpretation, writing. G.H.: literature search, study design, data interpretation, writing.

Competing interests. Throughout the paper's submission and editorial process Prof. G.H. was a member of the Royal Society Open Science Editorial Board. He had no involvement in the consideration of the manuscript for publication. Prof. Nick Pearce acted on this paper as Subject Editor for Science, Society and Policy. He is based at the University of Bath, as is one of the authors of the paper, but Prof. Pearce was not otherwise involved in the research. The other authors declare no conflicts of interest.

Funding. This research was supported by the Economic and Social Research Council.

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
