## [Peer Review File · Royal Society Open Science]

Review History

RSOS-210678.R0 (Original submission)

Review form: Reviewer 1

Is the manuscript scientifically sound in its present form?

No

Are the interpretations and conclusions justified by the results?

Yes

Is the language acceptable?

Yes

Do you have any ethical concerns with this paper?

No

Have you any concerns about statistical analyses in this paper?

No

Recommendation?

Major revision is needed (please make suggestions in comments)

Comments to the Author(s)

This paper describes a study which at two timepoints investigates COVID-19 related policy in the UK and its psychological antecedents. I generally like this research and it is a useful addition to the literature. Before being able to recommend acceptance of the paper, there are a few concerns I would like the authors to address.

First of all, I really like the theoretical question about personal versus general threat and how that can impact on acceptance of covid-related policy via psychological distance. However, how and why distance impacts on construal level in this context could have been explained a bit more I think, as this is the most interesting part of this study in my view.

Generally in regard to the intro I think there are several aims discussed in a very limited space. Interconnectivity of attitudes, heuristics, and psychological distance for example are all discussed, but how these theoretically relate in the present context is not always as clear as I think it could be. While I think the research addresses important questions, the intro could be improved as the elements discussed are not yet tied together convincingly. Why is the present study focusing on these three topics in conjunction? DO they relate and if so, how?

The second main point of concern is the Judging CoVID threat via response measure. This is a complex item: "I know that covid is a serious illness because we do't put lockdowns in place for other illnesses" and such complexity always brings about some ambiguity in terms of how to interpret the results. For example, if someone disagrees with the item, does that mean that they don't think covid is a serious illness? Or does it mean that they do not infer that from the lockdown? The argument is the latter, but my guess is the former plays a role as well, which makes it difficult to interpret the results. This could also explain part of the correlation of this measure with support for lockdown policies and future policies.

The third point is that across measurement points, the policy has changed considerably. As the temporal effects are key to the study, it would be good to pay some attention to that and (perhaps in the supplements) discuss these differences and how this may affect the results.

Review form: Reviewer 2

Is the manuscript scientifically sound in its present form?

Yes

Are the interpretations and conclusions justified by the results?

Yes

Is the language acceptable?

Yes

Do you have any ethical concerns with this paper?

No

Have you any concerns about statistical analyses in this paper?

No

Recommendation?

Major revision is needed (please make suggestions in comments)

Comments to the Author(s)

I genuinely enjoyed reading this manuscript. You identify a crucial missing piece in the current Covid discourse which is evident in just about every Western country. While there is plenty of

information about the 'what' (what people think, what experts say, etc.), there is far less understanding about the 'how'; how do people reach their conclusions? It is refreshing to see scholars such as yourself seeking to understand the reasons behind people's perceptions, as opposed to taking those opinions and restrictions at face value. Indeed, a mountain of research on risk perception suggests that risk-estimation is an imperfect process where availability and reputational cascades can derail judgments of lay people and experts alike (see Sunstein's work on availability cascades). Sacred values arising from uncertainty should be challenged. Therefore, I found your research questions to be of immense value. Your discussion is particularly thought-provoking, defeasible, and well-written.

A general concern I have about your paper in its current form, however, is that the methodological package should be a bit stronger, particularly given the weight of your question. Below are some questions and comments about your paper that I hope you'll find helpful for developing your ideas.

1. I found your methods description to be very limited and I urge you to provide far more information about your items, wording, anchors of each scale, etc. Provide all scales, items that you have developed and note anchors within each scale (as opposed to a generic statement). I assume that your figures / graphs provide all information that you have collected, but this is not clearly stated. It is not clear how many items you asked for each question, were they grouped together for analytical purposes, etc. When I went to your OSF file, I saw lots of other information that is completely absent from your write-up (e.g., children make me feel uninterested, something about a pub photo, and enjoyment of family time during the lockdown). I assume that information is from this other, larger study. However, in absence of any context, a reader may find your variable selection in this reported study to be cherry-picked and selected out of context. Please address this issue.

2. I encourage you to clearly state what this other study was about, what it measured, and then once you get to your methods, just focus on your actual study. Did you do anything with the data? Was that study published anything? Also, you seek to provide an alternative to polling. One limitation I would encourage you to mention is that your sample is not representative. I suppose polls typically rely on representative samples, but Prolific does not.

On a separate note, I wonder if it is an option to present Time 1 data only (and maybe T2 as supplement)? The relationships driving your core DVs will likely remain the same (i.e. belief that Covid-19 restrictions were necessary will predict restrictions). Again, I don't know what data you have at your disposal and what you are doing with it, but it seems like a fairly rich dataset worth examining further.

3. Your direct threat assessment was a bit ambiguous, though I understand it is quite commonly used for Covid research. If you are interested in pursuing this line of inquiry in the future, you could consider asking participants to estimate risks using some numerical indicators (e.g., How many people who contract Covid-19 end up in hospital?). I recommend the Brookings Institute report: <https://www.brookings.edu/research/how-misinformation-is-distorting-covid-policies-and-behaviors/>

4. The theoretical foundation outlined on pages 3 – 5 needs to be more refined and be more precise. The introduction covers anywhere from cognitive dissonance, hypocrisy, attitude contagion, heuristics, and construal theory. You may also (or in lieu) consider the theory on risk perceptions and false consensus (see Kuran & Sunstein availability cascades). Either way, your methods make it clear that you are really not testing all of these theories. Which one/s offer the most defensible foundation for your hypothesis? What is the foundational hypothesis that you seek to test? Your research questions (page 5) show that you are interested in several different

questions. How do they fit together? I would encourage you to either present a simple model depicting your vision, or for the purposes of this study, discuss why you focused on those DVs.

5. Be consistent in your terminology. You use terms 'direct threat' for methods and results, but then you shift to 'personal threat'. The latter seems to be a better descriptor of your construct.
6. You measure policy side-effects and you award some attention to this issue. However, theoretical background behind it is quite limited.

In summary, please provide a lot more information about your methods and make your theoretical foundation more precise. I sincerely wish you the best of luck in your future research. I hope you continue challenging the narrative and that you continue understanding the origins behind people's attitudes towards Covid.

Decision letter (RSOS-210678.R0)

Dear Dr Foad

The Editors assigned to your paper RSOS-210678 "The limitations of polling data in understanding public support for COVID-19 lockdown policies" have now received comments from reviewers and would like you to revise the paper in accordance with the reviewer comments and any comments from the Editors. Please note this decision does not guarantee eventual acceptance.

Please submit your revised manuscript and required files (see below) no later than 21 days from today's (ie 28-May-2021) date. Note: the ScholarOne system will 'lock' if submission of the revision is attempted 21 or more days after the deadline. If you do not think you will be able to meet this deadline please contact the editorial office immediately.

on behalf of Dr Julian Huppert (Associate Editor) and Nick Pearce (Subject Editor)
openscience@royalsociety.org

Associate Editor Comments to Author (Dr Julian Huppert):

Comments to the Author:

Thank you for sending this manuscript for review. Both reviewers believe that there is new and useful information in this manuscript, but that there is considerable work needed to improve the manuscript. In particular, the methodology should be clarified substantially, addressing the points both authors make.

Reviewer comments to Author:

Reviewer: 1

Comments to the Author(s)

This paper describes a study which at two timepoints investigates COVID-19 related policy in the UK and its psychological antecedents. I generally like this research and it is a useful addition to the literature. Before being able to recommend acceptance of the paper, there are a few concerns I would like the authors to address.

First of all, I really like the theoretical question about personal versus general threat and how that can impact on acceptance of covid-related policy via psychological distance. However, how and why distance impacts on construal level in this context could have been explained a bit more I think, as this is the most interesting part of this study in my view.

Generally in regard to the intro I think there are several aims discussed in a very limited space. Interconnectivity of attitudes, heuristics, and psychological distance for example are all discussed, but how these theoretically relate in the present context is not always as clear as I think it could be. While I think the research addresses important questions, the intro could be improved as the elements discussed are not yet tied together convincingly. Why is the present study focusing on these three topics in conjunction? DO they relate and if so, how?

The second main point of concern is the Judging CoVID threat via response measure. This is a complex item: "I know that covid is a serious illness because we do't put lockdowns in place for other illnesses" and such complexity always brings about some ambiguity in terms of how to interpret the results. For example, if someone disagrees with the item, does that mean that they don't think covid is a serious illness? Or does it mean that they do not infer that from the lockdown? The argument is the latter, but my guess is the former plays a role as well, which makes it difficult to interpret the results. This could also explain part of the correlation of this measure with support for lockdown policies and future policies.

The third point is that across measurement points, the policy has changed considerably. As the temporal effects are key to the study, it would be good to pay some attention to that and (perhaps in the supplements) discuss these differences and how this may affect the results.

Reviewer: 2

Comments to the Author(s)

I genuinely enjoyed reading this manuscript. You identify a crucial missing piece in the current Covid discourse which is evident in just about every Western country. While there is plenty of

information about the ‘what’ (what people think, what experts say, etc.), there is far less understanding about the ‘how’; how do people reach their conclusions? It is refreshing to see scholars such as yourself seeking to understand the reasons behind people’s perceptions, as opposed to taking those opinions and restrictions at face value. Indeed, a mountain of research on risk perception suggests that risk-estimation is an imperfect process where availability and reputational cascades can derail judgments of lay people and experts alike (see Sunstein’s work on availability cascades). Sacred values arising from uncertainty should be challenged. Therefore, I found your research questions to be of immense value. Your discussion is particularly thought-provoking, defeasible, and well-written.

A general concern I have about your paper in its current form, however, is that the methodological package should be a bit stronger, particularly given the weight of your question. Below are some questions and comments about your paper that I hope you’ll find helpful for developing your ideas.

1. I found your methods description to be very limited and I urge you to provide far more information about your items, wording, anchors of each scale, etc. Provide all scales, items that you have developed and note anchors within each scale (as opposed to a generic statement). I assume that your figures / graphs provide all information that you have collected, but this is not clearly stated. It is not clear how many items you asked for each question, were they grouped together for analytical purposes, etc. When I went to your OSF file, I saw lots of other information that is completely absent from your write-up (e.g., children make me feel uninterested, something about a pub photo, and enjoyment of family time during the lockdown). I assume that information is from this other, larger study. However, in absence of any context, a reader may find your variable selection in this reported study to be cherry-picked and selected out of context. Please address this issue.

2. I encourage you to clearly state what this other study was about, what it measured, and then once you get to your methods, just focus on your actual study. Did you do anything with the data? Was that study published anything? Also, you seek to provide an alternative to polling. One limitation I would encourage you to mention is that your sample is not representative. I suppose polls typically rely on representative samples, but Prolific does not.

On a separate note, I wonder if it is an option to present Time 1 data only (and maybe T2 as supplement)? The relationships driving your core DVs will likely remain the same (i.e. belief that Covid-19 restrictions were necessary will predict restrictions). Again, I don’t know what data you have at your disposal and what you are doing with it, but it seems like a fairly rich dataset worth examining further.

3. Your direct threat assessment was a bit ambiguous, though I understand it is quite commonly used for Covid research. If you are interested in pursuing this line of inquiry in the future, you could consider asking participants to estimate risks using some numerical indicators (e.g., How many people who contract Covid-19 end up in hospital?). I recommend the Brookings Institute report: <https://www.brookings.edu/research/how-misinformation-is-distorting-covid-policies-and-behaviors/>

4. The theoretical foundation outlined on pages 3 – 5 needs to be more refined and be more precise. The introduction covers anywhere from cognitive dissonance, hypocrisy, attitude contagion, heuristics, and construal theory. You may also (or in lieu) consider the theory on risk perceptions and false consensus (see Kuran & Sunstein availability cascades). Either way, your methods make it clear that you are really not testing all of these theories. Which one/s offer the most defensible foundation for your hypothesis? What is the foundational hypothesis that you seek to test? Your research questions (page 5) show that you are interested in several different

questions. How do they fit together? I would encourage you to either present a simple model depicting your vision, or for the purposes of this study, discuss why you focused on those DVs.

5. Be consistent in your terminology. You use terms 'direct threat' for methods and results, but then you shift to 'personal threat'. The latter seems to be a better descriptor of your construct.

6. You measure policy side-effects and you award some attention to this issue. However, theoretical background behind it is quite limited.

In summary, please provide a lot more information about your methods and make your theoretical foundation more precise. I sincerely wish you the best of luck in your future research. I hope you continue challenging the narrative and that you continue understanding the origins behind people's attitudes towards Covid.

===PREPARING YOUR MANUSCRIPT===

===PREPARING YOUR REVISION IN SCHOLARONE===

Author's Response to Decision Letter for (RSOS-210678.R0)

See Appendix A.

Decision letter (RSOS-210678.R1)

Dear Dr Foad,

It is a pleasure to accept your manuscript entitled "The limitations of polling data in understanding public support for COVID-19 lockdown policies" in its current form for publication in Royal Society Open Science.

COVID-19 rapid publication process:

We are taking steps to expedite the publication of research relevant to the pandemic. If you wish, you can opt to have your paper published as soon as it is ready, rather than waiting for it to be published the scheduled Wednesday.

This means your paper will not be included in the weekly media round-up which the Society sends to journalists ahead of publication. However, it will still appear in the COVID-19 Publishing Collection which journalists will be directed to each week (<https://royalsocietypublishing.org/topic/special-collections/novel-coronavirus-outbreak>).

If you wish to have your paper considered for immediate publication, or to discuss further, please notify openscience_proofs@royalsociety.org and press@royalsociety.org when you respond to this email.

You can expect to receive a proof of your article in the near future. Please contact the editorial office (openscience@royalsociety.org) and the production office (openscience_proofs@royalsociety.org) to let us know if you are likely to be away from e-mail contact – if you are going to be away, please nominate a co-author (if available) to manage the proofing process, and ensure they are copied into your email to the journal.

on behalf of Dr Julian Huppert (Associate Editor) and Nick Pearce (Subject Editor)
openscience@royalsociety.org

Appendix A

Dr. Julian Huppert (Associate Editor)

Prof. Nick Pearce (Subject Editor)

Dear Dr. Huppert and Prof. Pearce,

Thank you for handling our manuscript “The limitations of polling data in understanding public support for COVID-19 lockdown policies”. We very much appreciated the positive tone of the reviews and have made substantial revisions to the paper in light of the recommendations made by the reviewers. Thanks to their suggestions we think the manuscript is now much improved and ready for resubmission. Below is a point-by-point response, outlining how we have addressed each reviewer comment. We have quoted examples of our revisions from the manuscript in our responses, but please note that not every change is reported below, as some revisions related to more than one comment.

Reviewer 1:

Comment 1

This paper describes a study which at two timepoints investigates COVID-19 related policy in the UK and its psychological antecedents. I generally like this research and it is a useful addition to the literature. Before being able to recommend acceptance of the paper, there are a few concerns I would like the authors to address.

First of all, I really like the theoretical question about personal versus general threat and how that can impact on acceptance of covid-related policy via psychological distance. However, how and why distance impacts on construal level in this context could have been explained a bit more I think, as this is the most interesting part of this study in my view.

Generally in regard to the intro I think there are several aims discussed in a very limited space. Interconnectivity of attitudes, heuristics, and psychological distance for example are all discussed, but how these theoretically relate in the present context is not always as clear as I think it could be. While I think the research addresses important questions, the intro could be improved as the elements discussed are not yet tied together convincingly. Why is the present study focusing on these three topics in conjunction (sic)? DO they relate and if so, how?

We are glad our work is seen as a useful addition to the literature. We understand the reviewer’s desire for greater detail, and we have expanded upon our treatment of the theoretical relevance of attitudes, dissonance, and construal level theory in both the introduction and discussion, as well as including relevant theory on availability cascades (recommended by Reviewer 2).

e.g., p.4, “To achieve a dissonant state therefore requires simultaneously making more than one attitude or behaviour salient. If only one perspective is examined in isolation, people can support a particular action without having to address any of the possible negative effects of that action, as little or no dissonance will be elicited.”

e.g., p, 5 “Furthermore, heuristic cues can work at the collective level, formed via ‘availability cascades’, where informational cues (setting our beliefs in line with our perceptions of the beliefs of others), and reputational pressures (avoiding censure by following the beliefs of others) combine to

allow collective beliefs to form.¹⁵ In the face of an unprecedented and urgent crisis like the COVID-19 pandemic, cues and availability heuristics such as these are likely to be very powerful. ”

e.g., p. 14, “Two additional theoretical future directions are worth considering in light of our findings. First, we report a clear dissonance between overall support for lockdown and the cost-benefit analyses of associated policies. It seems reasonable to suggest that people find it easier to support lockdown policies within a positive frame (‘saving lives’) than within a negative frame (‘causing harm to children’). It would be interesting to examine further whether making both the policy and its side effects salient simultaneously arouses dissonance⁸, and, if so, the potential for consequent attitudinal and behavioural change.”

Specifically, we show how the threat of the pandemic is likely to sit more at the general and abstract level of construal for most people, and how our evidence supports this position.

e.g., p.6, “Because the pandemic is, by its very nature, a global problem, it seems plausible to suggest that general level construals are going to be highly important for people in their evaluation of the threat resulting from COVID-19.”

We also agree that the theoretical relation among attitudes, heuristics, and psychological distance could have been made clearer. Accordingly, we have also added sections to the introduction which show how the three questions we tackle in our research provide different but related perspectives on polling data outputs, and how the aforementioned theories can inform this perspective in an interconnected manner. In particular, we link attitudinal and dissonance research to the question of supporting lockdown whilst also considering its side effects; we link research on heuristics and cues to the question of judging the threat via the response; and we link research on construal level theory to the question of threat at a personal vs. general level. We also outline how those theories and questions have an overarching perspective – that of needing to examine attitudes from multiple perspectives.

e.g., pp.3-4, “Before describing relevant theoretical positions it is important to outline our overarching perspective. We wanted to understand further what underpins consistent polling data that show high support for lockdown policies. In particular, we sought to determine the information that people use to assess the threat of COVID-19, as well as how they see the cost-benefit trade-off of lockdown policies. Typically, the generally high levels of support for lockdown policies reported by polling companies have been contextualised as representing the view that the public perceives the intended benefits of lockdown as outweighing the costs of its side effects. However, these data only examine support from one angle, and cannot capture a range of views regarding the impacts of lockdown. Additionally, we suggest that polling support for lockdown has often been interpreted as evidence that this is what people consider appropriate from their personal perspective. We offer an alternative hypothesis: that asking about support for lockdown in isolation does not adequately capture the concerns people also have about lockdown side effects, and that when people are asked to judge the threat of COVID-19, they base their judgment on the policy response (i.e., shift from ‘normality’ to lockdown), rather than on their assessment of the threat to their personal circumstances. In order to test this hypothesis, we examined three important and related questions. First, does support for lockdown policies also mean an awareness that side-effects a) exist; and b) are acceptable? Second, do people use the magnitude of the response to guide their support for lockdown? Third, do people use the personal threat to themselves and close others, or a more general assessment of the threat, to guide their support for lockdown?”

Comment 2

The second main point of concern is the Judging CoVID threat via response measure. This is a complex item: “I know that covid is a serious illness because we do’t (sic) put lockdowns in place for other illnesses” and such complexity always brings about some ambiguity in terms of how to interpret the results. For example, if someone disagrees with the item, does that mean that they don’t think covid is a serious illness? Or does it mean that they do not infer that from the lockdown? The argument is the latter, but my guess is the former plays a role as well, which makes it difficult to interpret the results. This could also explain part of the correlation of this measure with support for lockdown policies and future policies.

We acknowledge the reviewer’s view that the cited item on the “judging COVID-19 threat via response” scale contains some methodological ambiguity. This point is discussed in greater detail within the revised discussion, where we have added a dedicated section to limitations and future directions. We believe that future qualitative and quantitative research would be useful to examine the use of policy response as a cue to threat severity, both in the context of COVID-19 and beyond.

e.g., p. 13, “Before concluding, we note some limitations to our findings and consider future directions that could address them. First, most of our measures were developed in response to the pandemic and therefore, whilst informed by relevant attitudinal theory, it would be valuable to extend our research with additional qualitative and quantitative work which seeks to understand further how people formed their attitudes towards lockdown policies. Our findings offer a timely insight into how people felt at the time, but such work could help clarify any ambiguity in examining how people judge this and other threats via policy responses, how people interpret threat at different levels of construal,¹⁶ and how people quantify comparable risks.³⁵ Similarly, our data suggest that it is much easier to support lockdown in general when it is framed as the solution to the problem, than when the potential harms of the policy are considered. However, further work is needed if we are to more accurately quantify exactly where public tolerance lies in terms of trade-offs between COVID-19 risks, lockdown risks, and also the positive aspects of normal social interaction.”

However, we also still think this measure provides crucial information. First, we wish to note that the items combine to form a reliable single component which captures different aspects of the extent to which participants used the response as a guide to the threat, so we are not reliant on any single item for our findings. Second, our sample (aligned with polling data) is generally very strongly in support of lockdown policies, therefore, whilst it is true that there are two different reasons to disagree with the cited item, we are confident that stronger agreement with this overall measure represents participants who have used the lockdown response as a guide to the severity of the threat, more than it captures a general assessment of the severity of COVID-19 as a serious illness.

Comment 3

The third point is that across measurement points, the policy has changed considerably. As the temporal effects are key to the study, it would be good to pay some attention to that and (perhaps in the supplements) discuss these differences and how this may affect the results.

This is an apt point, and we have included a section in the discussion to note how lockdown policies have changed over time.

e.g., pp. 13-14, "Second, across the six-month period, many policies were constantly changing. Our measures were designed to examine attitudes independent of current policy, but longitudinal data could further test the relationship between the introduction of a policy and support for that policy, both within and outside the COVID-19 context."

We have also added analyses in the supplemental materials reporting where change has or has not occurred in the measures that are not already reported in the manuscript (see Figure S1). The two-phase structure of the data is useful as it allows us to see whether support for each policy is stable or dynamic, how this relates to concurrent polling data (which we refer to in the results), and whether the relationships we found in phase 1 were still reliable six months later (which they all were). Because of the fast-moving nature of relevant policy, our measures were designed to be as independent as possible to the actual policy of the time, hence our examination of initial lockdown support on March 23rd 2020, and our assessment of future support for different policies, all of which were not in place at phase 1, but some of which were in place at phase 2. This framing was designed to allow us to examine participants' perceptions of past and future support, without referring directly to current policy. We therefore note where we think temporal change is relevant, but in terms of our three main questions, all the findings replicate at phase 2, giving us confidence that these mechanisms are both reliable, and not just a result of policy at any particular timepoint.

Reviewer 2:

Comment 1

I genuinely enjoyed reading this manuscript. You identify a crucial missing piece in the current Covid discourse which is evident in just about every Western country. While there is plenty of information about the 'what' (what people think, what experts say, etc.), there is far less understanding about the 'how'; how do people reach their conclusions? It is refreshing to see scholars such as yourself seeking to understand the reasons behind people's perceptions, as opposed to taking those opinions and restrictions at face value. Indeed, a mountain of research on risk perception suggests that risk-estimation is an imperfect process where availability and reputational cascades can derail judgments of lay people and experts alike (see Sunstein's work on availability cascades). Sacred values arising from uncertainty should be challenged. Therefore, I found your research questions to be of immense value. Your discussion is particularly thought-provoking, defensible, and well-written.

A general concern I have about your paper in its current form, however, is that the methodological package should be a bit stronger, particularly given the weight of your question. Below are some questions and comments about your paper that I hope you'll find helpful for developing your ideas.

1. I found your methods description to be very limited and I urge you to provide far more information about your items, wording, anchors of each scale, etc. Provide all scales, items that you have developed and note anchors within each scale (as opposed to a generic statement). I assume that your figures/graphs provide all information that you have collected, but this is not clearly stated. It is not clear how many items you asked for each question, were they grouped together for analytical purposes, etc. When I went to your OSF file, I saw lots of other information that is completely absent from your write-up (e.g., children make me feel uninterested, something about a pub photo, and

enjoyment of family time during the lockdown). I assume that information is from this other, larger study. However, in absence of any context, a reader may find your variable selection in this reported study to be cherry-picked and selected out of context. Please address this issue.

We are glad the reviewer sees our work as identifying a crucial missing part of Covid discourse and thank them for their recommendation of examining the research on availability cascades (see comment 4). We have expanded our statements regarding how these measures are taken from a larger project examining several perspectives on COVID-19 attitudes and other important psychological variables, and clearly stated that these variables were designed specifically for addressing the questions raised in this manuscript in terms of what polling data might be missing.

e.g., p.7, “The measures reported below were designed specifically to address the question of how people formed their attitudes towards lockdown policies and have not been published in any form anywhere else. Future articles are planned to outline other psychological processes of interest from these datasets. A full list of items collected for each phase are available (see OSF link after Discussion).”

This is the first paper we have written based on these datasets, but we hope to publish more findings in the future on other components of the data that are designed to address very different questions (e.g., the role of attitudes towards children, personal values, and lockdown experiences). We hope this satisfies any concerns that these measures were in any way selected out of context.

We agree that the methodological description could have been more thorough and accordingly we have significantly increased the coverage of the methods both in the manuscript and via supplemental materials.

e.g., p.8, To reduce items into reliable scales, we adopted the same process for all measures. First, we entered the items for each measure into a principal components analysis (PCA), with Varimax rotation used when multiple components were present. We retained any components with an Eigenvalue greater than 1 and excluded any remaining items. Furthermore, to be retained, all items required a component loading of at least .60, and no cross loading with other components greater than .50. The PCA analyses showing all retained and excluded items are available in supplemental materials (Tables S1-S5), as well as the question framing and all scale anchor points. All measures reported below were designed for the purposes of this research, although they were informed by items being used by polling companies and government policies at the time of data collection. All the PCAs were run on phase 1 data and all the scales developed then showed good reliability at both phase 1 and phase 2 (see Table 1).”

As stated above, all items, anchors, and question frames for the measures are now in supplemental materials, along with the principal components analyses (PCAs) for every measure, which outline how the scales were developed. The full list of items and question wording for every item at both phases of our larger project have also been put onto OSF. We are grateful for the opportunity to make this part of the research clearer, as it has allowed us to make a couple of small changes to the measures in order that all the PCAs use identical criteria for item inclusion, which we think is the most effective way of communicating how the measures were formed. The only meaningful change is that our “support for initial COVID-19 restrictions” is now made of two components (previously one combined scale) representing items with very high agreement, and items of lower agreement. Previously we had combined these into one component as they theoretically covered a good range of measures and had excellent internal validity. But we can see the value in reporting these separately;

doing so fits our criteria for item selection consistently across measures, and the additional analyses add further support to our hypothesis working at different levels of agreement. Additionally, one further item has been excluded from each of the “social relations” and “society” components of the policy side effects measure. Crucially, all of these changes do not affect the strength of our findings, and in fact provide further reassurance that our findings are robust.

Comment 2

2. I encourage you to clearly state what this other study was about, what it measured, and then once you get to your methods, just focus on your actual study. Did you do anything with the data? Was that study published anything? Also, you seek to provide an alternative to polling. One limitation I would encourage you to mention is that your sample is not representative. I suppose polls typically rely on representative samples, but Prolific does not.

Please see the previous comment which covers the first part of the reviewer’s comment. In terms of the representativeness of the sample, whilst it is true that Prolific does not ordinarily provide a representative sample in the same way polling companies do, it is still a public sample which provides a good spread in terms of gender, age, SES and education. The sample is somewhat skewed to left-wing and remain-voting participants when it comes to political views, but we examined the correlations between political viewpoint, Brexit viewpoint, and all our measures at phase 1; there were no strong relationships (all $r_s < .17$, all $p_s > .01$, see Table S6 in supplemental materials). We can therefore be confident that our findings would be equally powerful in a standard representative sample. Despite these reassuring data, we do note the utility of using representative samples in our discussion.

e.g., p.14, “Third, whilst we used a public sample from a commonly used participant panel (Prolific), which offers significant heterogeneity in terms of participant gender, age, education, and socio-economic status, the sample does have some political skew (towards left-wing and remain-voting participants). Nonetheless, we still had good sample sizes from centrists, right-wing, and leave-voting participants and did not find any strong relationships between political viewpoint and any of our measures (see Table S6) so we are confident the mechanisms we identify would translate into a more typically representative sample. However, given the comparisons with polling data, we do acknowledge this difference in sampling methodology.”

Comment 3

3. Your direct threat assessment was a bit ambiguous, though I understand it is quite commonly used for Covid research. If you are interested in pursuing this line of inquiry in the future, you could consider asking participants to estimate risks using some numerical indicators (e.g., How many people who contract Covid-19 end up in hospital?). I recommend the Brookings Institute report: <https://www.brookings.edu/research/how-misinformation-is-distorting-covid-policies-and-behaviors/>

As the reviewer notes, the direct (personal) threat assessment items have been commonly used in COVID research, but we agree that in line with other future directions now included in the discussion, further work would be useful to learn more about how participants perceived, and potentially quantified, the threat of COVID-19.

e.g., p.13, “Before concluding, we note some limitations to our findings and consider future directions that could address them. First, most of our measures were developed in response to the pandemic and therefore, whilst informed by relevant attitudinal theory, it would be valuable to extend our research with additional qualitative and quantitative work which seeks to understand further how people formed their attitudes towards lockdown policies. Our findings offer a timely insight into how people felt at the time, but such work could help clarify any ambiguity in examining how people judge this and other threats via policy responses, how people interpret threat at different levels of construal,¹⁶ and how people quantify comparable risks.³⁵”

We thank the reviewer for the recommended report and cite this in the discussion accordingly (also in the above quote from p. 13).

Comment 4

4. The theoretical foundation outlined on pages 3 – 5 needs to be more refined and be more precise. The introduction covers anywhere from cognitive dissonance, hypocrisy, attitude contagion, heuristics, and construal theory. You may also (or in lieu) consider the theory on risk perceptions and false consensus (see Kuran & Sunstein availability cascades). Either way, your methods make it clear that you are really not testing all of these theories. Which one/s offer the most defensible foundation for your hypothesis? What is the foundational hypothesis that you seek to test? Your research questions (page 5) show that you are interested in several different questions. How do they fit together? I would encourage you to either present a simple model depicting your vision, or for the purposes of this study, discuss why you focused on those DVs.

This point very much aligns with Comment 1 from Reviewer 1, so please see our response to that comment. We think it is necessary to cite a range of relevant psychological theories, as they were all used to inform our theoretical and methodological development, but we agree that the integration with our research questions could have been clearer so we have addressed that issue in the introduction. The three research questions are introduced earlier in the introduction, and the following three paragraphs then show how our theoretical foundation informs each one (again, outlined in more detail in Comment 1). On top of that we thank the Reviewer for their recommendation of Kuran & Sunstein’s work on availability cascades, which we have also incorporated into the manuscript.

e.g., p.5, “Furthermore, heuristic cues can work at the collective level, formed via ‘availability cascades’, where informational cues (setting our beliefs in line with our perceptions of the beliefs of others), and reputational pressures (avoiding censure by following the beliefs of others) combine to allow collective beliefs to form.¹⁵”

e.g., p.12, Furthermore, polling data have been shown to be susceptible to socially desirable responding,²⁵ which aligns with theory on availability cascades,¹⁵ and could also contribute to exaggerating the true level of support for lockdown policies”

In sum, we think the manuscript now offers a much clearer relationship between the three main questions we seek to address, and the theoretical underpinning to those questions.

Comment 5

5. Be consistent in your terminology. You use terms 'direct threat' for methods and results, but then you shift to 'personal threat'. The latter seems to be a better descriptor of your construct.

We agree that 'personal threat' is a better descriptor and hence now use that term consistently throughout the manuscript.

Comment 6

6. You measure policy side-effects and you award some attention to this issue. However, theoretical background behind it is quite limited.

In line with the other comments on theoretical integration, we have added further theoretical background on ambivalence which outlines the utility in examining attitudes from multiple perspectives which directly informs our first research question. We have also added a comment in the discussion that integrates research that specifically examines how people make causal attributions in relation to side effects which we think would be a fascinating future direction in relation to COVID-19.

e.g., p. 14, "Second, extant research on side effects shows that different types of side effect attract different attributions at the level of the individual.³⁶ It would be valuable to extend our understanding of how the side effects of lockdown policies are seen in terms of causation, and how this might impact upon people's tolerance for such effects."

To conclude, we are very grateful for the constructive feedback provided. We have carefully considered each and every point and believe that addressing this feedback has improved the manuscript significantly. We hope you will now find it acceptable for publication in *Royal Society Open Science*.

Kind regards,

Colin Foad, Lorraine Whitmarsh, Paul Hanel, and Geoff Haddock